# Prolonging the circulatory half-life of C1 esterase inhibitor via albumin fusion

**Sangavi Sivananthan**[1], **Varsha Bhakta**[1,2], **Negin Chaechi Tehrani**[1], **William P. Sheffield** [1,2] *

1 Department of Pathology and Molecular Medicine, McMaster University, Hamilton, Ontario, Canada,
2 Canadian Blood Services, Innovation and Portfolio Management, Hamilton, Ontario, Canada

* sheffiel@mcmaster.ca

**Data Availability Statement:** All relevant data are within the paper and its Supporting Information files.

## Abstract

Hereditary Angioedema (HAE) is an autosomal dominant disease characterized by episodic swelling, arising from genetic deficiency in C1-esterase inhibitor (C1INH), a regulator of several proteases including activated Plasma kallikrein (Pka). Many existing C1INH treatments exhibit short circulatory half-lives, precluding prophylactic use. Hexahistidine-tagged truncated C1INH (trC1INH lacking residues 1–97) with Mutated N-linked Glycosylation Sites N216Q/N231Q/N330Q ($H_6$-trC1INH(MGS)), its murine serum albumin (MSA) fusion variant ($H_6$-trC1INH(MGS)-MSA), and $H_6$-MSA were expressed in *Pichia pastoris* and purified via nickel-chelate chromatography. Following intravenous injection in mice, the mean terminal half-life of $H_6$-trC1INH(MGS)-MSA was significantly increased versus that of $H_6$-trC1INH (MGS), by 3-fold, while remaining ~35% less than that of $H_6$-MSA. The extended half-life was achieved with minimal, but significant, reduction in the mean second order rate constant of Pka inhibition of $H_6$-trC1INH(MGS)-MSA by 33% relative to that of $H_6$-trC1INH(MGS). Our results validate albumin fusion as a viable strategy for half-life extension of a natural inhibitor and suggest that $H_6$-trC1INH(MGS)-MSA is worthy of investigation in a murine model of HAE.

## Introduction

Hereditary Angioedema (HAE) is an autosomal dominant disease characterized by an acute episodic swelling in the stomach, face, limbs, and throat; if the airways are involved HAE attacks can be life-threatening [1, 2]. There are two subtypes of HAE. Type 1 is characterized by both reduced C1 inhibitor (C1INH) antigen and activity levels, while type 2 is marked by normal antigenic but reduced activity C1INH levels. Both stem from mutations in *SERPING1* [3]. C1INH is a glycoprotein serine protease inhibitor belonging to the serpin superfamily [4, 5]. The multifaceted activities of C1INH include roles in vascular permeability, anti-inflammatory function, coagulation, and fibrinolysis [6–9].

C1INH is the largest serpin (110kDa) and is hyperglycosylated with 10 sites of N-glycosylation and 24 sites of O-glycosylation [10, 11]. Most of these sites lie in a 100 amino acid N-terminal extension not found in other members of the serpin superfamily [5], with the exception

**Funding:** This study was made possible through a Canadian Blood Services (CBS) Graduate Fellowship Program award to SS, and via a CBS Discovery Research Grant DRG2023-WS to WS. The funders had no role in study design, data collection and analysis, decision to publish, or preparation of the manuscript.

**Competing interests:** The authors have declared that no competing interests exist.

of 3 N-linked glycosylation sites, N216, N231, and N330 [11] (numbered according to mature protein sequence). Several research groups have demonstrated that recombinant C1INH retains wild-type-like inhibitory activity when its N-terminal 97 residues were deleted [12–14], but, at least in the baculovirus/insect cell expression system, combining this truncation with the triple mutation N216Q/N231Q/N330Q to eliminate glycosylation abrogated detectable expression [13].

C1INH plays a pivotal role in inhibiting various proteases, including C1s in the complement system and plasma kallikrein (Pka) [15]. Following cleavage, Pka catalyzes the cleavage of high molecular weight kininogen, liberating bradykinin. Bradykinin controls fluid balance between the circulation and tissues via the B2 receptor, whose binding results in endothelial cell contraction, vasodilation, and nitric oxide production [16]. Bradykinin levels correlate with HAE attack severity and are higher in affected limbs than unaffected limbs in the same HAE patients during attacks [17, 18]. These observations, combined with the clinical efficacy of replacement therapy with either C1INH concentrates or drugs inhibiting Pka, support a causative role for unregulated bradykinin activity in HAE [19–22].

Over the last decade, the clinical management of HAE has expanded from treating attacks acutely to preventing attacks via prophylactic treatment [23]. Fewer drugs are approved for HAE prophylaxis than for acute treatment; for example, the only existing recombinant C1INH in clinical use, conestat alfa, is unsuitable for prophylactic use due to its short half-life [24]. Theoretically, prophylactic coverage for HAE patients would be easier to achieve with extended half-life (EHL) products, as has been demonstrated with protein drugs used to treat patients with hemophilia [25]. One approach to an EHL for HAE treatment would be a C1INH-albumin fusion protein. Fusing a therapeutic protein to albumin typically extends its plasma residency time, becoming more like that of albumin; plasma-derived human serum albumin (HSA) exhibits a plasma half-life of 19 days [26]. We hypothesized that albumin fusion would not impair C1INH-mediated inhibition of Pka and would extend its circulatory half-life *in vivo*. To facilitate an EHL strategy for C1INH, we simplified C1INH by removal of its N-terminal 97 amino acids, mutation of its serpin domain N-linked glycosylation sites and addition of an N-terminal hexahistidine tag to facilitate purification. We report that the inhibitory properties of simplified, truncated, non-glycosylated C1INH were largely unaffected by genetic fusion to mouse serum albumin (MSA) and that this fusion protein acquired an extended plasma residency time in mice in vivo approaching that of hexahistidine-tagged recombinant MSA.

## Methods

### Expression and purification of $H_6$-trC1INH(MGS), $H_6$-MSA, and $H_6$-trC1INH(MGS)-MSA fusion protein in *P. pastoris*

Clone Manager 10 (Sci Ed Software LLC) was used for design and oligonucleotides, or other synthetic DNA sequences (gBlock$^{TM}$) were purchased from Integrated DNA Technologies (IDT). All restriction and DNA modification enzymes were bought from Thermo Fisher Scientific (Waltham, Massachusetts, United States) (XhoI, catalogue number (cat. #) FD0694, EcoRI, cat. # FD0274, AccIII, cat. # FD0534). A hexahistidine-tagged truncated C1INH$_{98-478}$ with Mutated Glycosylation Sites (MGS; N216Q/N231Q/N330Q) ($H_6$-trC1INH(MGS)) was designed and the resulting 1192 base pair (bp) $H_6$-C1INH gBlock$^{TM}$ product was restricted with XhoI and EcoRI and inserted between these sites in pPICZ9ssamp [27, 28] to yield pPICZ9ss$H_6$-trC1INH(MGS). An MSA cDNA was modified for expression via PCR, as directed by sense oligonucleotide 5'-TCTCTCGAGA AAAGACATCA CCATCACCAT CACGAAGCAC ACAAGAGTGA GATCGCC-3' and antisense oligonucleotide 5'-CCTAGGGAAT

TCCTAGGCTA AGGCGTCTTT GCATCTAGTG AC-3' and catalyzed by heat-stable polymerase Phusion under conditions directed by its manufacturer (Thermo Fisher Scientific, cat. # XXXXXX). The resulting amplicon was restricted with XhoI and EcoRI and inserted between these sites in pPICZ9ssamp to form pPICZ9ssH$_6$-MSA. Fusion construct H$_6$-trC1INH(MGS)-MSA, was assembled by ligating a 1206 bp H$_6$-trC1INH(MGS)-(G$_4$S)$_3$ gBlock$^{TM}$ product restricted with XhoI and AccIII, and a 1797 bp (G$_4$S)$_3$-MSA gBlock$^{TM}$ product restricted with AccIII and EcoRI. These fragments were then inserted between the Xho1 and EcoRI sites of pPICZ9ssamp vector to yield pPICZ9ssH$_6$-trC1INH(MGS)-MSA. All plasmid candidates were verified by Sanger dideoxy sequencing of the relevant open reading frames by the McMaster University Genomics Facility. Purified plasmid DNA was linearized with appropriate restriction enzymes and were used to transform *Pichia pastoris (P. pastoris)* strain X-33 to Zeocin resistance. The transformed cell lines were cultured, induced with methanol, and the secreted proteins H$_6$-trC1INH(MGS), H$_6$-trC1INH(MGS)-MSA, and H$_6$-MSA were purified from conditioned media using nickel-chelate affinity chromatography, as previously described [29].

## Protease inhibition assays with H$_6$-trC1INH(MGS) and H$_6$-trC1INH (MGS)-MSA

Second-order rate constants (k$_2$) and stoichiometries of inhibition (SI) were determined using chromogenic assays as previously described [30]. Pka was purchased from Enzyme Research Laboratories (South Bend, IN, USA) (cat. # 1303) and complement component 1, s subcomponent (C1s), activated, two chain form, was bought from Sigma Aldrich (Oakville, ON, Canada) (cat. # 204879-250UG). The velocity of amidolysis of chromogenic substrate S2302 (DiaPharma, West Chester, OH, USA) (cat.# 82034039) by Pka or of chromogenic substrate Pefachrome™-C1E (CH$_3$CO- benzyloxycarbonyl Lys-Gly-Arg-para-nitroanilide monoacetate) (DiaPharma) (cat. # DPG08703) by C1s was assessed in 96-well flat-bottom microtiter plates (Immulon IV, Thermo Fisher Scientific) (cat. # 1424563) at 37˚C in PPNE buffer (20 mM sodium phosphate pH 7.4, 0.1% (w/vol) polyethylene glycol 8000, 100 mM sodium chloride, and 0.1 mM disodium ethylenediaminetetraacetic acid). The measurements were performed at a wavelength of 405 nm using an Elx808 Absorbance Microplate Reader (Biotek, Winooski, VT, USA). For the determination of k$_2$, reactions included 10 nM Pka (Enzyme Research, South Bend, USA) or C1s (Millipore Sigma, Massachusetts, USA) and 200 nM pdC1INH (Athens Research and Technology, Athens, Georgia, USA) (cat. # 16-16-031509) or C1INH recombinant variants, incubated at 30-second intervals and halted with 100 µM S2302 or 400 µM Pefachrome™-C1E chromogenic substrate. To ascertain the SI, reactions included 10 nM Pka and 10–50 nM pdC1INH/C1INH recombinant variants, incubated for 2-hours and stopped with 100 µM S2302.

## Electrophoretic analysis of H$_6$-trC1INH(MGS) and Pka interaction

The electrophoretic profile of the interaction between pdC1INH/H$_6$-trC1INH(MGS) with Pka was visualized on 10% (w/vol) sodium dodecyl sulfate (SDS)-polyacrylamide gels (SDS-PAGE, under reducing conditions. Reactions involved a final concentration of 5 µM pdC1INH/H$_6$-trC1INH(MGS) and 1 µM Pka in PPNE buffer and were conducted at 37˚C for 5-minutes. Reactions were then terminated by adding 1/3 the reaction volume of concentrated 4x SDS-PAGE sample buffer, and samples containing the entire reaction volume were subjected to electrophoresis. Gels were stained with Coomassie Brilliant Blue as described previously [31]. Polyacrylamide gels were imaged by scanning using a model XR GelDoc system (Bio-Rad Laboratories, Mississauga, ON, Canada) and nitrocellulose immunoblots prepared as described [30] were imaged by scanning using a model Azure 280 system (Azure Biosystems, CA, USA), generating Tagged Image File (TIF) format outputs.

## Detection of C1INH binding to kallikrein by modified ELISA

An enzyme-linked immunosorbent assay (ELISA) was employed to detect $H_6$-trC1INH (MGS)-MSA: Pka complexes, as described, with modifications [31]. In 96-well flat-bottom microtiter plates, 250 ng of Pka (in 0.1 ml of PBS) was coated overnight. Subsequently, unbound Pka was removed, and the wells were washed with PBS-T (136 mM sodium chloride, 2.7 mM potassium chloride, 10 mM disodium phosphate, 1.8 mM monopotassium phosphate, and 0.1% Tween-20). $H_6$-trC1INH(MGS)-MSA (1 μg or 5 μg) or pdC1INH (500 ng or 1 μg) were diluted in 1% bovine serum albumin (BSA) in PBS-T and incubated in the Pka-coated 96-well flat bottom microtiter plates for 1-hour. Wells were subsequently washed with modified PBS-T buffers containing 1 M sodium chloride and 1% Tween-20 for the remaining washing steps. C1INH-related protein binding to immobilized Pka was detected with 100 ng per well of goat anti-human C1INH antibody conjugated to horseradish peroxidase (HRP) (Affinity Biologics, Ancaster, Canada) (cat. # GACINH-HRP); colour generation was monitored using 3,3',5,5'-Tetramethylbenzidine (TMB) substrate solution (Thermo Fisher Scientific) (cat. # 34021), 0.1 ml/well, and was stopped by adding an equal volume of 2M sulphuric acid after 5 minutes, prior to quantification using an Elx808 Absorbance Microplate Reader.

## *In vivo* clearance and protein detection in mice

All *in vivo* experiments in this study used CD1 mice (Charles River, Wilmington, MA, USA) and were approved by the Animal Research Ethics Board of the Faculty of Health Sciences, McMaster University, via Animal Utilization Protocol 20-02-06. The ARRIVE guidelines (https://arriveguidelines.org/arrive-guidelines) were followed (specifically the ARRIVE Essential 10). Mice were typically 1–2 months old. All mice were acclimatized to their surroundings in the Central Animal Facility of the Faculty of Health Sciences of McMaster University for at least one week after arrival. Mice were always maintained under anesthesia during the procedure, using 3% isoflurane. For clearance studies, mice were intravenously injected with either 50 μg $H_6$-trC1INH(MGS), 50 μg $H_6$-MSA, or 100 μg $H_6$-trC1INH(MGS)-MSA, each in 0.2 ml of saline solution. The dose was selected in order to administer equimolar doses of the three proteins to be compared, allowing for the greater molecular weight of $H_6$-trC1INH(MGS)-MSA than the other two proteins, to ensure that 1% of the initial dose could be detected in plasma samples using ELISA, and to not exceed ~20% of the plasma concentration of C1INH in wild-type mice, expected to be ~0.25 mg/ml [6]. Serial blood samples were collected from the tail over time, and recombinant proteins remaining in plasma derived from these samples were quantified by ELISA with minor modifications [31]. Groups of six mice comprising three male and three female mice were used in all cases, with sample size selection being based on results of previous studies [29, 32]. Three groups were initially compared, those receiving $H_6$-trC1INH(MGS) or $H_6$-trC1INH(MGS)-MSA or $H_6$-MSA, the latter serving as a control. Subsequently two additional groups of six mice each receiving $H_6$-trC1INH(MGS) or $H_6$-trC1INH(MGS)-MSA were compared. A total of 30 mice were used in this study, weighing 31 ± 4 g (mean ± SD). Neither randomization nor blinding were employed. All animals were included in the analysis; there were no exclusions. After the final blood sample was drawn, mice were euthanized by cervical dislocation under anesthetic cover. To capture $H_6$-trC1INH (MGS) and $H_6$-trC1INH(MGS)-MSA, 0.1 ml of 1 μg/ml goat anti-human C1INH (Affinity Biologics, Ancaster, ON, Canada) (cat. # GACINH) was coated, per well, on a 96-well flat-bottom microtiter plate, overnight. Bound C1INH proteins were detected using 0.3 μg/ml goat anti-human C1INH HRP-conjugated antibody in 1% BSA in PBS, with colour detection as described above. Plasma-derived C1INH was used to construct a standard curve linear between 0.5 and 20 ng/ml. For detecting $H_6$-MSA, plasma samples diluted 1:1 with PBS were

incubated with 500ng of Biotin-SP-conjugated affinity-purified Rabbit Anti-His tag antibody (Jackson ImmunoResearch Laboratories, PA, USA) (cat. # 300-065-240) for 1-hour before transfer to streptavidin-coated plates (Thermo Fisher Scientific) (cat. # 15120). Bound antibody-MSA complexes were detected with 0.25 μg/ml HRP-conjugated goat polyclonal anti-MSA antibodies (Affinity Biologicals) (cat. # AB19195) in 1% BSA in PBS, with colour generation as described above. Purified $H_6$-trC1INH(MGS)-MSA was used to construct a standard curve linear between 0.5 ng/ml and 20 ng/ml. Terminal half-lives were determined using a two-compartment model and curve-stripping as previously described [33]. The area under the curve was calculated via GraphPad Prism version 9.0 (Insight Partners, New York, NY, USA).

## Statistical analysis and significance testing

Data are presented as the mean ± the standard deviation (SD). Statistical analysis was conducted using GraphPad Prism version 9.0. A p-value of $< 0.05$ was indicative of statistical significance. For multiple comparisons, data were assessed using One-way ANOVA with post-tests.

## Results

### Novel recombinant proteins

In this study, three recombinant proteins were designed, expressed, and purified, using a *P. pastoris* expression system, as previously described. In each case, recombinant minigenes were under the control of the methanol-inducible alcohol oxidase 1 (AOX1) promoter, and recombinant proteins were secreted into the conditioned media by the *Saccharomyces cerevisiae* yeast prepro-α factor 80 amino acid cleavable prepro signal sequence, terminating in the Kex2/Ste13 protease cleavage site (SLEKR↓EA) [34]. Each protein thus shared the Glu-Ala dipeptide followed by a hexahistidine tag on its N-terminus; their predicted primary sequences are shown in S1 Appendix. Fig 1A shows the three recombinant proteins $H_6$-trC1INH(MGS), $H_6$-trC1INH(MGS)-MSA, and $H_6$-MSA, with reference to full-length plasma-derived C1INH (pdC1INH).

The electrophoretic profile of the recombinant proteins is shown in Fig 1B, following nickel chelate affinity chromatography. $H_6$-trC1INH(MGS) was purified to homogeneity and contained a single polypeptide of 43 kDa, matching its predicted molecular weight of 43,697 Da. Similarly, $H_6$-trC1INH(MGS)-MSA exhibited a single polypeptide band of approximately 110 kDa, consistent with its predicted molecular weight of 110, 315 Da. Both C1INH variant proteins reacted with antibodies specific to human C1INH and hexahistidine; in addition, $H_6$-trC1INH(MGS)-MSA acquired immunoreactivity with antibodies specific for MSA. $H_6$-MSA preparations comprised a major polypeptide of 67 kDa consistent with the predicted molecular weight of 66,713 Da, as well as minor polypeptide comprising approximately 10% of the total protein of approximately 55 kDa. Both the major and minor species reacted not only with anti-MSA antibodies on immunoblots but also with anti-hexahistidine, indicating that the latter was intact on its N-terminus but was missing C-terminal residues.

### Characterization of fused and unfused truncated, non-glycosylated C1INH proteins

The kinetics of protease inhibition by $H_6$-trC1INH(MGS) and $H_6$-trC1INH(MGS)-MSA were compared to that mediated by pdC1INIH, as shown in Table 1. Neither truncation, prevention of N-linked glycosylation, nor hexahistidine tagging appeared to impact inhibition of C1s, as the $k_2$ for C1s inhibition did not differ from pdC1INH. Adding albumin to the list of changes

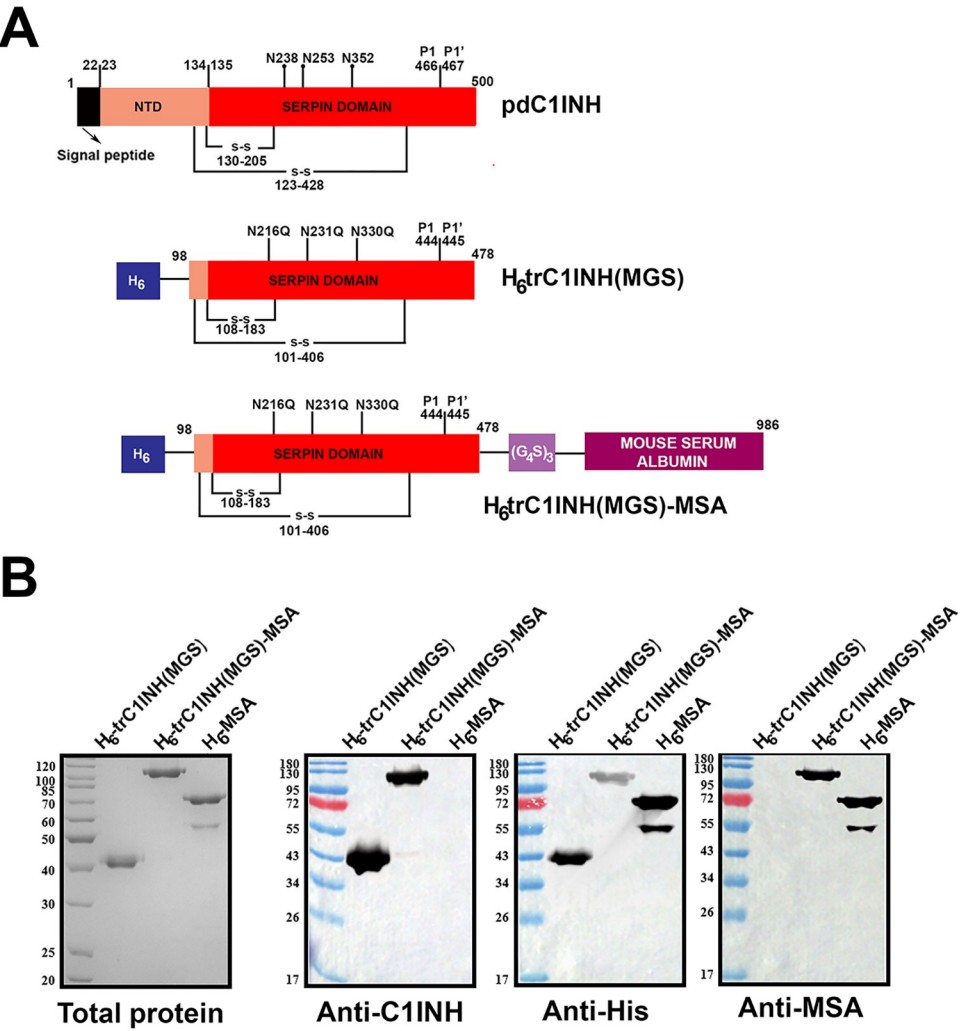

**Fig 1. C1-esterase inhibitor (C1INH) recombinant proteins.** A) Schematic diagram of pdC1INH and recombinant proteins. Numbers relate to C1INH mature protein except as otherwise specified. Proteins are named at right. B) Gel and immunoblot analysis of purified $H_6$-trC1INH(MGS), $H_6$-trC1INH(MGS)-MSA, and $H_6$-MSA. Total protein amounts of 1 μg were electrophoresed on 10% SDS-PAGE gels that were stained with Coomassie Brilliant Blue (left) or decorated with specific antibodies identified below the panels (left centre, right centre, and right).

**Table 1. Pharmacokinetic characterization of activity of C1INH recombinant proteins.**

| Protein name | $k_2$ versus Pka (X $10^4$ $M^{-1}s^{-1}$) | $k_2$ versus C1s (X $10^4$ $M^{-1}s^{-1}$) | SI versus Pka |
|---|---|---|---|
| pdC1INH | 2.3 ± 0.2*** | 5.7 ± 0.5 | 3.3 ± 0.06***^^^ |
| $H_6$-trC1INH(MGS) | 2.01 ± 0.07*** | 5 ± 0.5 | 5.3 ± 0.3 |
| $H_6$-trC1INH(MGS)-MSA | 1.3 ± 0.08 | 5.1 ± 0.4 | 6.9 ± 0.1 |

Results are the mean of 5 determinations (SI or $k_2$ for kallikrein), ± SD, or 3 determinations ($k_2$ for C1s).

***, $p < 0.001$ versus $H_6$-trC1INH(MGS)-MSA.

^^^, $p < 0.001$ versus $H_6$-trC1INH(MGS).

similarly did not impact the inhibition of C1s by $H_6$-trC1INH(MGS)-MSA. In contrast, small but significant reductions in mean $k_2$ for Pka inhibition were noted for both unfused and fused trC1INH(MGS) proteins, of 13.4% for $H_6$-trC1INH(MGS) and 45.2% for $H_6$-trC1INH (MGS)-MSA versus pdC1INH. Mean SI values were significantly increased for $H_6$-trC1INH (MGS) and $H_6$-trC1INH(MGS)-MSA compared to pdC1INH, by factors of 1.6 and 2.1, respectively, but did not differ significantly between each other. The results indicate small but significant reductions in both the rate and efficiency of inhibition for the variant recombinant C1INH proteins versus their pdC1INH counterpart.

Inhibitors of the serpin class of protease inhibitors typically form denaturation-resistant, SDS-stable complexes [35]. Inhibitory complexes between Pka and plasma-derived inhibitors were first examined by SDS-PAGE. As shown in Fig 2Ai, commercial purified Pka was comprised of multiple polypeptide chains when electrophoresed under reducing conditions, including a catalytic light chain of ~37 kDa. This polypeptide chain was almost fully converted into novel, C1INH-dependent high molecular weight complexes when Pka was reacted with an excess of either pdC1INH or $H_6$-trC1INH(MGS), of 147 kDa or 80 kDa, respectively. In contrast, when the same reaction was carried out with Pka and excess $H_6$-trC1INH(MGS)-MSA (Fig 2Aii), no higher molecular weight serpin-enzyme complex was formed, although

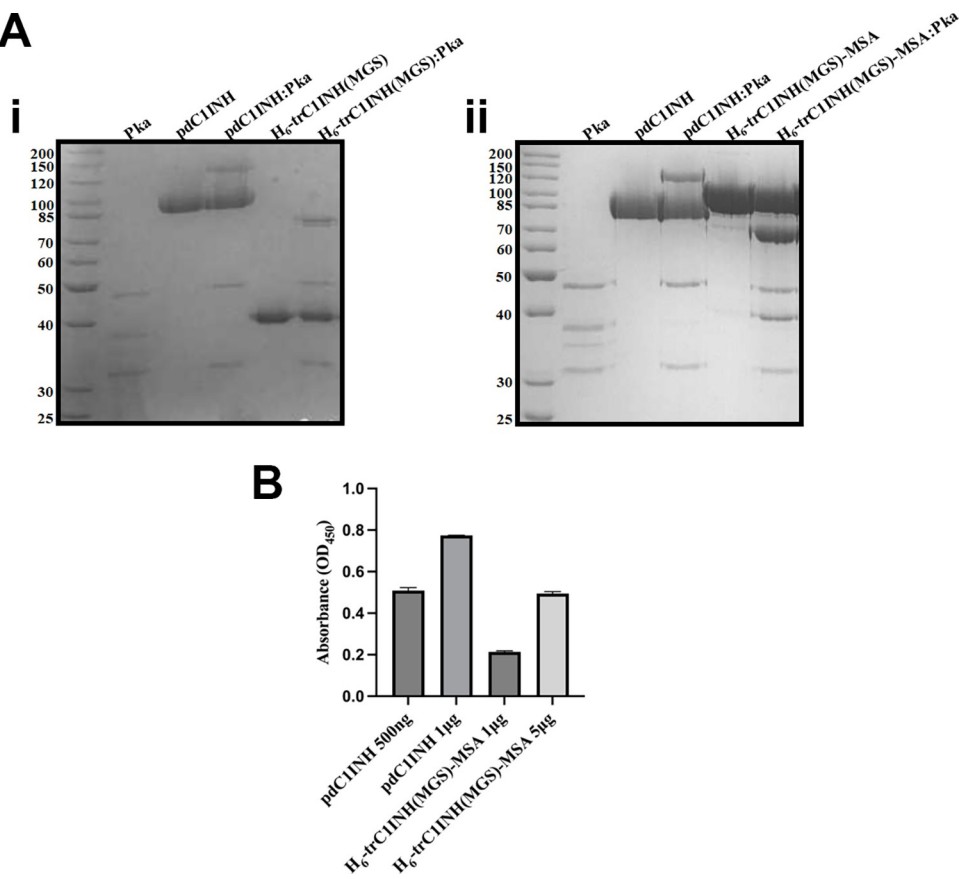

**Fig 2. Complex characterization of C1INH recombinant proteins.** A) i. Gel-based assays of complex formation. 10% SDS-PAGE gels are shown electrophoresed under reducing conditions and stained with Coomassie Brilliant Blue. Serpins (pdC1INH, $H_6$-trC1INH(MGS), or $H_6$-trC1INH(MGS)-MSA) were reacted with Pka at a 5:1 molar ratio for 5-minutes at 37°C. B) Graph of binding assay between pdC1INH:Pka and $H_6$-trC1INH(MGS)-MSA:Pka at varying ratios (500 ng-1 μg) with PBS-T washing condition of 1M NaCl and 1% Tween-20. The mean and SD of 3 determinations is shown.

the Pka catalytic light chain appeared to be consumed to a similar extent as in the reaction with $H_6$-trC1INH(MGS). Two new polypeptide chains were generated from $H_6$-trC1INH(MGS)-MSA on exposure to Pka, of ~70 kDa and ~40 kDa.

To reconcile the apparent discrepancy of a substantial $k_2$ for Pka inhibition with the inability to detect SDS-stable complexes between $H_6$-trC1INH(MGS)-MSA and Pka by SDS-PAGE, we sought independent verification of complex formation. Active Pka was immobilized on microtiter plate wells, incubated with pdC1INH or $H_6$-trC1INH(MGS)-MSA, washed under stringent high-salt, high detergent conditions, and captured C1INH antigenic material was detected with C1INH-specific antibodies. As shown in Fig 2B, bound C1INH was detected when either pdC1INH or $H_6$-trC1INH(MGS)-MSA was reacted with immobilized Pka, in a dose-dependent manner, although the signal was higher with pdC1INH than with $H_6$-trC1INH(MGS)-MSA. These results, like the $k_2$ findings, are suggestive of tightly bound inhibitory complex formation between Pka and $H_6$-trC1INH(MGS)-MSA even if the complexes are not SDS-stable.

### *In vivo* clearance of recombinant proteins

$H_6$-trC1INH(MGS), $H_6$-trC1INH(MGS)-MSA, and $H_6$-MSA proteins were separately injected into groups of mice. The average relative recovery in serial plasma samples is shown in Fig 3A (log-linear plot) and Fig 3B (linear-linear plot), where the profile of $H_6$-trC1INH(MGS) in mice exhibited a distinct and more rapidly cleared trajectory from that of $H_6$-trC1INH(MGS)-MSA or $H_6$-MSA, within minutes of injection. After 1 hour, only $27 \pm 10\%$ of the initial dose of $H_6$-trC1INH(MGS) remained in the circulation, which was significantly lower than the residual levels of $H_6$-trC1INH(MGS)-MSA or $H_6$-MSA ($90 \pm 4\%$ and $95 \pm 3\%$, respectively) (Fig 3C). By 4 hours post-injection, only $2.2 \pm 0.5\%$ of the initial dose of $H_6$-trC1INH(MGS) remained in the circulation, significantly less than $H_6$-trC1INH(MGS)-MSA or $H_6$-MSA ($48 \pm 10\%$ and $60 \pm 8\%$, respectively) (Fig 3D). While residual levels of $H_6$-trC1INH(MGS)-MSA and $H_6$-MSA did not exhibit a significant difference at 1 hour, the difference became significant by 4 hours post-injection.

Our initial design of the in vivo clearance experiment was limited by local animal research ethics requirements which restricted the number of blood samples that could be taken in a 24-hour period. For $H_6$-trC1INH(MGS) that design covered >95% of the total clearance curve, but <50% of that of the less rapidly cleared MSA-containing proteins (see Fig 3A–3D). More extrapolation was therefore required to estimate the half-life of those proteins than for $H_6$-trC1INH(MGS). Accordingly, an additional experiment was performed in which identical doses of $H_6$-trC1INH(MGS)-MSA and $H_6$- MSA as previously employed were administered to additional groups of 6 mice each, and blood samples were obtained 16–28 hours after injection. As shown in Fig 3E, these results supported the initial, shorter experiment, and showed that substantial quantities of $H_6$-trC1INH(MGS)-MSA ($19 \pm 1\%$) and $H_6$- MSA ($22 \pm 1\%$) 18 and 28 hours after injection, respectively.

### Pharmacokinetic analysis

The use of a two-compartment model to resolve clearance curves allowed the calculation of terminal half-lives, alongside the area under the curve (AUC) for the three proteins, are presented in Table 2. Fusion to albumin significantly extended the mean terminal half-life of C1INH by 3.0-fold. However, $H_6$-trC1INH(MGS)-MSA still exhibited a 1.5-fold more rapid clearance compared to $H_6$-MSA. Furthermore, there was a significant 3.8-fold increase in the observed AUC for $H_6$-trC1INH(MGS)-MSA versus $H_6$-trC1INH(MGS), while remaining 1.1-fold less than the AUC of $H_6$-MSA. The results suggest that albumin fusion significantly

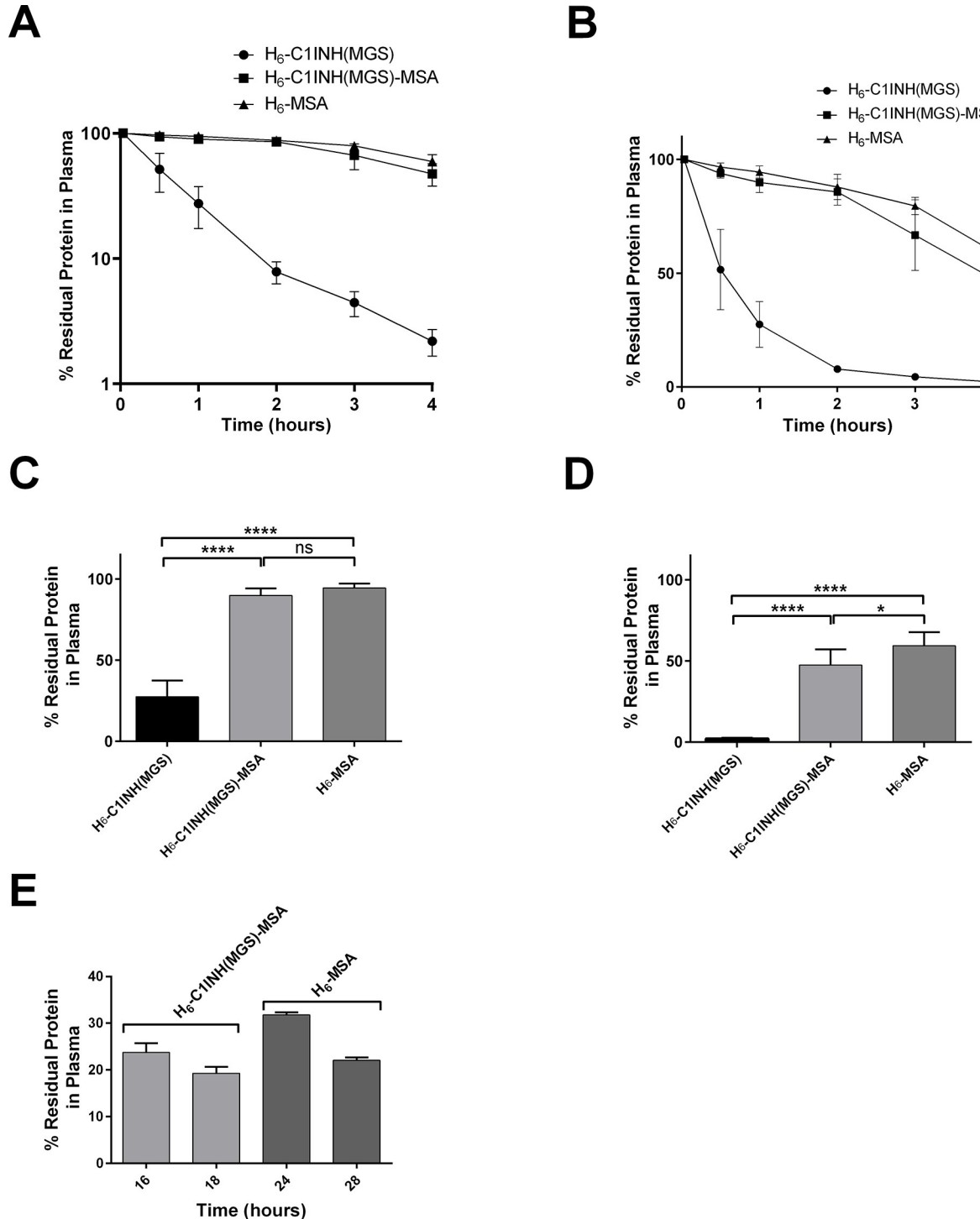

**Fig 3. Protein clearance of H₆-trC1INH(MGS), H₆-trC1INH(MGS)-MSA, and H₆-MSA.** A) Logarithmic-linear plot of $H_6$-trC1INH (MGS), $H_6$-trC1INH(MGS)-MSA, and $H_6$-MSA percentage residual protein in plasma versus time in hours post-injection. B) Linear plot of $H_6$-trC1INH(MGS), $H_6$-trC1INH(MGS)-MSA, and $H_6$-MSA percentage residual protein in plasma versus time in hours post-injection. C) Percentage residual protein in plasma at 1-hour post-injection. D) Percentage residual protein in plasma at 4-hours post-injection. E) Percentage residual protein in plasma at the times, in hours, specified on the x axis, following injection of $H_6$-trC1INH(MGS)-MSA (light grey bars) or $H_6$-MSA (dark grey bars). The mean of 6 determinations ± SD is shown in all cases. *, $p < 0.05$, ****, $p < 0.0001$ for statistical comparisons indicated by horizontal bars (ns denotes non-significant).

Table 2. $H_6$-trC1INH(MGS), $H_6$-trC1INH(MGS)-MSA, $H_6$-MSA pharmacokinetic analysis of clearance.

| Protein name | Terminal half-life (hours) | Area Under the observed Curve (AUC; %-hours) |
|---|---|---|
| $H_6$-trC1INH(MGS) | $5 \pm 2$ | $80 \pm 20$ |
| $H_6$-trC1INH(MGS)-MSA | $14 \pm 3^*$ | $310 \pm 20^{***}$ |
| $H_6$-MSA | $21 \pm 8^{***}$ | $340 \pm 8^{***}$ |

Results are the mean of 6 determinations, ± SD.

*, $p < 0.05$

***, $p < 0.001$ versus $H_6$-trC1INH(MGS).

extended the plasma residency time of $H_6$-trC1INH(MGS)-MSA compared to $H_6$-trC1INH (MGS), as well as the terminal half-life, supporting our hypothesis, although the extension fell short of achieving the longer values observed for $H_6$-MSA. No effect of sex on terminal half-life was noted for any of the three proteins; when half-life values were grouped by sex for each protein, no significant differences were noted (as indicated by p values of 0.1 to 0.7 by Mann-Whitney non-parametric test). Similarly, no effect of body weight was noted for any of the three proteins; when half-life values were divided into heavier half versus lighter half per protein per group, no significant differences were noted (as indicated by p values of 0.4 to 0.99 by Mann-Whitney non-parametric test).

## Discussion

The chief findings of this study were that the inhibitory properties of C1INH were maintained in a truncated, non-glycosylated variant, that these properties were largely maintained when this variant was fused to serum albumin, and that the fusion protein remained in the circulation of mice after intravenous injection for considerably longer than its unfused equivalent. Although the results supported our initial hypothesis, the extension in plasma residency fell short of that exhibited by recombinant albumin alone.

Unfused $H_6$-trC1INH(MGS) left the circulation rapidly after injection, as expected for a protein of 43 kDa, below the glomerular filtration limit. Such proteins are expected to be filtered in the glomerular structures of the kidneys, and not re-adsorbed in the renal tubules [36]. Consistent with this expectation, the clearance curve for $H_6$-trC1INH(MGS) diverged from those of $H_6$-trC1INH(MGS)-MSA or $H_6$-MSA at the earliest sampled time point. Both plasma residency time, as quantified by the observed AUC, and terminal catabolism, as quantified by the terminal half-life, increased 3- to 4-fold when MSA was fused to $H_6$-trC1INH (MGS). $H_6$-trC1INH(MGS)-MSA approached, but did not reach, the long plasma residency and slow rate of terminal clearance of $H_6$-MSA, achieving mean levels of 92% of the AUC and 66% of the terminal half life of that hexahistidine-tagged albumin. The mean terminal half-life of $H_6$-MSA, 21.4 hours, was less than that reported for plasma-derived MSA in previous studies (27.7 hours [37]– 35 hours [38]), a difference potentially arising either from hexahistidine tagging, recombinant production, or methodological differences such as protein radioiodination in previous studies. In this work we elected to follow the fusion proteins without modification via ELISA of plasma samples, in part due to initial observations that pdC1NH lost the ability to form SDS-stable complexes with Pka after iodination. Our results were nonetheless internally consistent and showed a significant extension of half-life and reduction in clearance rate for $H_6$-trC1INH(MGS)-MSA versus $H_6$-trC1INH(MGS). The improved pharmacokinetic profile was likely obtained by the combined mechanisms of avoidance of renal filtration and the conferring of recycling via murine FcRn [39].

Fusion to albumin generally increases the plasma half-life of fused proteins, but it can interfere with the biological function of the attached protein, to varying extents. Miyakawa et al. reported that fusion of interferon γ to MSA increased the observed clearance AUC of the fusion protein 10-fold versus unfused interferon γ, but at the cost of the loss of ~99% of its biological activity [40]. Less dramatic losses in activity were reported by our laboratory in fusing the Kunitz Protease Inhibitor (KPI) domain of Protease Nexin 2 to HSA, where the affinity of KPI for its target protease FXIa significantly declined 2.1-fold with fusion [41]. Similarly, fusing FIX to HSA resulted in substantial losses in specific activity compared to unfused recombinant FIX of up to 100-fold, unless a cleavable spacer was used that could liberate FIX from HSA [42]. In this study the rate of C1s inhibition by $H_6$-trC1INH(MGS) was unaffected by MSA fusion and the rate of Pka inhibition was reduced 1.8-fold. These outcomes likely arose from favourable interactions between the $H_6$-trC1INH(MGS) moiety of the fusion protein and the target proteases, unimpeded by MSA, which was designed to be distanced from that moiety by a substantive triplicated $Gly_4Ser$ spacer.

Previous investigators have established that truncated, glycosylated recombinant C1 inhibitor retained the inhibitory capacity of full-length C1INH against several protease targets. Coutinho et al. deleted the N-terminal 97 residues of C1INH and showed, qualitatively, similar inhibitory activity of the deletion mutant as wild-type C1INH produced in cultured COS-1 cells [12]. Bos et al. expressed the same truncation mutant (Δ97) in *P. pastoris* and reported comparable association rate constants differing by less than two-fold for wild-type and truncated C1INH for C1s, Pka, and FXIIa [14]. Rossi et al. made the Δ97 truncation using baculovirus expression vectors expressed in cultured Sf21 insect cells, finding equivalent complex formation with C1s on gel analysis both for truncated and hexahistidine-tagged C1inhΔ97-ht and for untagged C1inhΔ97, and for truncated variants with mutated N216 and/or N231 sites; truncated variants with mutated N216, N231, and N330 sites failed to be expressed [13]. Our results contrasted with those of Rossi et al. in that we were able to obtain enough hexahistidine-tagged, truncated, non-glycosylated $H_6$-trC1INH(MGS) to determine quantitatively its identical kinetics of C1s inhibition and its similar (85%) kinetics of Pka inhibition to its plasma-derived counterpart. The discordance suggests that glycosylation on N330 is obligatory only for expression in Sf21 insect cells [13] and not for eukaryotic cells in general.

While many studies of albumin fusion proteins employ HSA as a fusion partner rather than a species-matched albumin, it is known that the serum half-life of HSA fusion proteins can be underestimated in mice. This underestimation arises because the binding affinity of HSA to murine FcRn is about 10-fold weaker than its binding to human FcRn [43]. Moreover, HSA-conjugated drugs persist longer in the circulation of transgenic mice expressing human FcRn than in wild-type mice [44]. Site-specific conjugation of superfolder green fluorescent protein (sfGFP) to MSA or HSA resulted in a significantly longer mean half-life for the sfGFP-MSA conjugate (27.7 hours) than for sfGFP-HSA (16.3 hours) [45]. In addition, use of MSA as the fusion partner could make it easier to determine proof-of-concept for the utility of $H_6$-trC1INH(MGS)-MSA in knockout mice genetically deficient in C1INH [6].

Purified plasma-derived C1INH concentrates have been reported to have mean half-lives in HAE patients ranging from 32.7 [46] to 74.1 hours [47]. These values might seem longer than those reported in this study for both $H_6$-trC1INH(MGS) and $H_6$-trC1INH(MGS)MSA were it not for the principle of allometric scaling, which holds that drug or protein clearance is proportional to the body weight of an animal or human[48]. The mean terminal catabolic half-life of HSA in humans is 19 days [26]. Extrapolating from our results in mice, we speculate that the half-life of a $H_6$-trC1INH(MGS)-HSA fusion protein would be significantly longer than that of plasma-derived C1INH concentrates, although it is unlikely that the full extension of half-life to 19 days would be achieved.

Although $H_6$-trC1INH(MGS)-MSA inhibited Pka with similar kinetics to either unfused $H_6$-trC1INH(MGS) or pdC1INH, in contrast to those proteins, the fusion protein did not form detectable SDS-stable complexes with Pka. Our failure to detect covalent serpin-protease complexes with this combination of serpin and protease may have arisen for several reasons. Firstly, our choice of reaction conditions could have been responsible; greater excesses of $H_6$-trC1INH(MGS)-MSA over Pka might have been required because of its mean SI of 6.8, versus 5.3 for $H_6$-trC1INH(MGS) (although these values did not differ statistically) and 3.2 for pdC1INH; others have estimated the latter value, for pdC1INH, at approximately 2 [14]. Secondly, it is possible that the fusion protein formed a strongly stabilized non-covalent inhibitory complex. There are precedents for such unusual variant serpins. Ciaccia et al. did not detect a covalent complex between the L444R variant of heparin cofactor II and α-thrombin in the presence of heparin, despite strong inhibition of activity [49]. Similarly, Rossi et al. described the lack of covalent complex formation between Complement Component 1, r subcomponent-like protein (C1rLP) and C1inhΔ97, despite strong inhibition of C1rLP activity [13]. Finally, relatively few fusion proteins involving serpins have been described in the literature. Alpha-1 antitrypsin (AAT), the most abundant serpin in human plasma, loses the ability to inhibit its natural target, neutrophil elastase, when fused to the N-terminus of IgG1 Fc [50]. In contrast, AAT M358R variant retains function when fused, via its N-terminus, to the C-terminus of bacteriophage T7 coat protein 10, as evidenced by its formation of SDS-stable complexes with thrombin [51]. Thus, $H_6$-trC1INH(MGS) retains potent anti-Pka activity but may inhibit its target via an atypical mechanism.

The rate of inhibition of Pka by C1INH is surprisingly low compared to those achieved by other serpins in regulating other proteases [35]. Nevertheless, the susceptibility of C1INH-deficient individuals to HAE and their successful treatment with pdC1INH concentrates show that it is high enough to maintain control of bradykinin generation. In this study we showed that fusion protein $H_6$-trC1INH(MGS)-MSA gains extended half-life and maintains most of rate of Pka inhibition exhibited by pdC1INH. Future studies in C1INH knockout mice will be required to determine if this profile is sufficiently favourable to warrant continued development via substitution of the MSA moiety by HSA and investigation in mice with human FcRn, or if protein engineering to re-direct a different serpin towards Pka inhibition at higher rates, like the approach taken by de Maat et al. [52] would be a superior strategy.

## Supporting information

**S1 Appendix.** (1) Protein sequences and lengths. The primary sequence of all recombinant proteins employed in this study are shown, along with the length in amino acids, the molecular weight in Daltons, and the calculated isoelectric point. (2) Raw data shown in Table 1. (3) Raw data shown in Fig 2B. (3) Raw data shown in Fig 3A–3D. (4) Raw data shown in Fig 3E. (5) Raw data shown in Table 2.
(DOCX)

**S2 Appendix. This file contains uncropped original images of all gels and blots shown in this study.**
(PDF)

## Author Contributions

**Conceptualization:** William P. Sheffield.

**Data curation:** Sangavi Sivananthan, Varsha Bhakta, Negin Chaechi Tehrani.

**Formal analysis:** Sangavi Sivananthan, William P. Sheffield.

**Funding acquisition:** William P. Sheffield.

**Investigation:** Sangavi Sivananthan, Varsha Bhakta, Negin Chaechi Tehrani.

**Methodology:** Sangavi Sivananthan, Varsha Bhakta, Negin Chaechi Tehrani.

**Project administration:** William P. Sheffield.

**Resources:** William P. Sheffield.

**Supervision:** William P. Sheffield.

**Writing – original draft:** Sangavi Sivananthan, William P. Sheffield.

**Writing – review & editing:** Sangavi Sivananthan, Varsha Bhakta, Negin Chaechi Tehrani, William P. Sheffield.

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
