## [Decision Letter · Decision Letter 0]

10 Apr 2024

PONE-D-24-10397Prolonging the circulatory half-life of C1 esterase inhibitor via albumin fusionPLOS ONE

Dear Dr. Sheffield,

Thank you for submitting your manuscript to PLOS ONE. After careful consideration, we feel that it has merit but does not fully meet PLOS ONE’s publication criteria as it currently stands. Therefore, we invite you to submit a revised version of the manuscript that addresses the points raised during the review process.

Both the reviewers have optimistic and raised some critical issues in constructive manner. Authors need to update their literature survey with more pertinent studies and paralells are needed to be drawn comparing the merits of the findings.Authors also need to revise methods while furnishing details of reagent / biologicals sources. There is a missing information regarding sex and weight having effect on outcome and dosing flexiblity according to weight. 

All in all this study has sufficient vigour for publication in PlosOne but requires critical revision.

We look forward to receiving your revised manuscript.

Kind regards,

Yash Gupta, Ph.D.

Academic Editor

PLOS ONE

Journal Requirements:

"This study was made possible through a Canadian Blood Services (CBS) Graduate Fellowship Program award to SS, and via a CBS Discovery Research Grant DRG2023-WS to WPS."

"This study was made possible through a Canadian Blood Services (CBS) Graduate Fellowship Program award to SS, and via a CBS Discovery Research Grant DRG2023-WS to WS. Due to the funding of the discovery research program at CBS by a contribution agreement with Health Canada, a department of the federal government of Canada, this publication must contain, as a condition of funding, the statement “The views expressed herein do not necessarily represent the views of the federal government of Canada.” "

"This study was made possible through a Canadian Blood Services (CBS) Graduate Fellowship Program award to SS, and via a CBS Discovery Research Grant DRG2023-WS to WPS."     

Reviewers' comments:

Reviewer's Responses to Questions

**Comments to the Author**

1. Is the manuscript technically sound, and do the data support the conclusions?

Reviewer #1: Yes

Reviewer #2: Yes

2. Has the statistical analysis been performed appropriately and rigorously? 

Reviewer #1: Yes

Reviewer #2: Yes

3. Have the authors made all data underlying the findings in their manuscript fully available?

Reviewer #1: Yes

Reviewer #2: Yes

4. Is the manuscript presented in an intelligible fashion and written in standard English?

Reviewer #1: Yes

Reviewer #2: Yes

5. Review Comments to the Author

Reviewer #1: Journal: PLOS ONE

Manuscript Number: PONE-D-24-10397

Article Type: Research Article

Title: Prolonging the circulatory half-life of C1 esterase inhibitor via albumin fusion

Recommendation: Publish after minor revisions are noted.

Comments:

The manuscript by Sivananthan and colleagues described the Hereditary Angioedema (HAE) an autosomal dominant episodic swelling disease occur in various body parts due to deficiency in C1-esterase inhibitor (C1INH), a regulator of several proteases eg. Plasma Kallikrein (Pka). Till date, many C1INH treatments have been employed to prevent this disease yet exhibit short circulatory half-life. In this study, they designed, expressed and purified recombinant proteins H6-trC1INH(MGS), H6-trC1INH(MGS)-MSA, and H6-MSA using Pichia pastoris. They found the significant extended circulatory half-life of recombinant H6-trC1INH(MGS)-MSA compared to H6-trC1INH(MGS).

Author also tried to perform vivo study using the CDI mice in which they injected the N-glycosylated C1INH recombinant protein with /or without albumin fusion, separately. They found the mean terminal half-life of H6-trC1INH(MGS)-MSA was 3-fold higher with reduction in Pka inhibition than that of wildtype H6-trC1INH(MGS). Suggesting that H6-trC1INH(MGS)-MSA discovery is advantage in mice model of HRE.

Overall, this an interesting manuscript showing extended circulatory half-life of N-glycosylated C1INH recombinant protein via albumin fusion.

Comment 1: Please add the catalog number for all purchased reagents. For examples: Antibodies, Pka, Complement component.

Comment 2: What is the rationale for dose determination of N-glycosylated C1INH recombinant proteins injected to mice model?

Comment 3: Please see the page 15, line 287,H6-trC1INH(MGS)-MSA or H6-MSA (90±4% and 95%±3%, respectively. please delete % from 95%.

Comment 4: Please tale a look on Y-axis scale of figure 3A. It looks to weird. There should be 50% residual protein in the middle of y axis scale instead of 10%. Please correct it accordingly.

Reviewer #2: In this work, Sivanathan et al. suggest that albusion prolongs the circulatory half-life of C1 esterase inhibitor. The title is suitable and precise, and the abstract adequately summarises work presented. The work is clear and straightforward and presented well. The data presented, structured, and the manuscript is well written with targeted experiment.

Points

1. Authors showed that albumin fusion extends the half-life of the inhibitor, but studies have already shown the half-life has extended up to several days.

The recent manuscript by Martinez-Saguer et al. suggests that the residual half-life of C1-esterase inhibitor (C1INH) treatment is up to 144 hours. (doi.org/10.1016/j.jacig.2023.100178.)

2. The authors should provide further valid points, such as why albumin fusion has better therapeutic potential than already available options, which have a shelf life for several days.

3. Do authors measure the protein level beyond the 4-hour time point?

4. Do authors observe any significant differences in experimental outcomes between male and female mice?

5. The authors mentioned that mice weighed between 25 and 35 g, the range of weight difference seems broad. Does the weight of mice have any difference in clearance of the protein?

6. In the methods section “In vivo clearance and protein detection in mice” did the authors used separate standards for H6-MSA and 6-trC1INH(MGS) or used the same standards (Purified H6-trC1INH(MGS)-MSA) for all the residual protein quantification.

6. PLOS authors have the option to publish the peer review history of their article (what does this mean?). If published, this will include your full peer review and any attached files.

Reviewer #1: **Yes: **Brijesh Singh Chauhan

Reviewer #2: No

---

## [Author Response · Author response to Decision Letter 0]

28 May 2024

We have attached a Response to Reviewers file and also reproduce the document below:

Response to Reviewers of PONE-D-24-10397

Our submission was invited for revision 2024-04-10 if we addressed all points raised during the review process. We have done so. Our responses are organized under “Journal Requirements”, “Reviewer 1”, and “Reviewer 2”.

Journal Requirements

1. The editor directed us to ensure that the manuscript meets PLOS ONE’s style requirements. We have verified that we have done so in our revised submission.

2. The editor asked us to enlarge our financial disclosure. We have done so in our cover letter, which now states the financial disclosure as “This study was made possible through a Canadian Blood Services (CBS) Graduate Fellowship Program award to SS, and via a CBS Discovery Research Grant DRG2023-WS to WS. The funders had no role in study design, data collection and analysis, decision to publish, or preparation of the manuscript.”

3. The Editor pointed out that we had repeated funding information in the Acknowledgements Section of the manuscript and that this error needed to be corrected. We have deleted the Acknowledgements section from the revised manuscript. In addition, we have further enlarged the Funding Statement to contain the previously misplaced text, “Due to the funding of the discovery research program at CBS by a contribution agreement with Health Canada, a department of the federal government of Canada, this publication must contain, as a condition of funding, the statement, “The views herein do not necessarily represent the views of the federal government of Canada.””

4. The editor pointed out that we needed to provide all uncropped and unadjusted images underlying all gel and blot results. These have now been provided in accordance with journal policies as S2 Appendix in Supporting Information.

5. The editor asked that our full ethics statement be provided. We have verified that our Animal Utilization Protocol details are provided in the Methods, including the full name of the issuing body.

6. The editor reminded us to include captions for Supporting Information files at the end of the manuscript and to update in-text citations where appropriate. We have added captions for the two Supporting Information files (S1 Appendix and S2 Appendix). In-text citations have been doublechecked for updating. 

7. The editor concluded by asking that we review the reference list and to note any changes to the reference list. We have carried out this review. We report that we have added 3 references to address reviewers’ concerns, as described in detail in the response to reviewers’ comments below (see Reviewer 2, Point 1). We added an additional citation of a pre-existing reference in providing additional information in the text concerning dose selection (see Reviewer 1, Point 2).

Reviewer 1

1. The reviewer requested that catalog numbers be provided for all purchased reagents. These numbers have now been provided. All purchased reagents, a total of 16 items, are now identified by catalogue number. 

2. The reviewer asked for the rationale for the dose determination of the C1INH recombinant proteins that we injected into mice. This information has now been provided in the Methods section, where we now state, “The dose was selected in order to give equimolar doses of the three proteins to be compared, allowing for the greater molecular weight of H6-trC1INH(MGS)-MSA than the other two proteins, to ensure that 1% of the initial dose could be detected in plasma samples using ELISA, and to not exceed ~20% of the plasma concentration of C1INH in wild-type mice, expected to be ~0.25 mg/ml [6].” 

3. The reviewer found a typographical error and requested its correction. We have corrected “95%±3%” to read “95±3%”.

4. The reviewer commented that the Y-axis scale of Figure 3A looked odd, and that there should be 50% residual protein in the middle of the Y axis rather than 10%. The reason for this feature of Figure 3A is that it is a log-linear plot, with the Y axis being logarithmic in scale. Because protein clearance is typically described by a biphasic natural decay function, we thought this was the best way to present the data. However, to respect the reviewer’s perspective, we have added a linear-linear plot of the same data as a new panel in the revised figure, Figure 3B (rev).

Reviewer 2

1. The reviewer commented that our data supported an extension of C1INH half-life associated with albumin fusion but pointed out that a recent study by Martinez-Saguer et al. reported a half-life of several days for clinical grade C1INH concentrates in HAE patients. We have added a section to the Discussion referencing the study and addressing this point. Briefly, therapeutic protein clearance follows the principle of allometric scaling, in which animals with low body weights (or total body surface area) exhibit faster protein clearance than humans. Martinez-Saguer reported a half-life of 74.1 ± 19 hours (mean ± SD) for purified human plasma-derived C1INH in 20 HAE patients treated with a dose of 20 IU/kg. The half-life of human albumin in humans is 19 days. We therefore hypothesize that a C1INH-albumin fusion protein would persist in the circulation for much longer than current products like that employed by Martinez-Saguer. Moreover, due to allometry, the half-life of albumin in mice is only one day. Our data are predictive of a half-life in humans of a C1INH-albumin fusion protein intermediate between that of current products and albumin. The Discussion has been modified to make these points clearer. In so doing we added 3 references: to the Martinez-Saguer et al. study, to a similar study examining the pharmacokinetics of another plasma-derived C1INH concentrate in HAE patients, and to an article about allometry, i.e. dose extrapolations between animals and humans.

2. The reviewer’s second point was a direction to explain why albumin fusion has good therapeutic potential. We have done so, as explained in Point 1.

3. The reviewer asked if we measured the protein level beyond the 4-hour time point. In our original submission, we did not. However, we have added additional data to the revised manuscript. For H6-MSA and H6-trC1INH-MSA, we verified that substantial quantities of the proteins remained in the murine circulation at longer time points of 16 – 28 hours post-injection, recognizing that the original experiment required much more extrapolation to characterize the half-lives of these longer-lived proteins than H6-trC1INH(MGS), which was >95% removed from the circulation within the original four-hour period. These additional data are now shown in new panel E of revised Figure 3. 

4. The reviewer asked if we observed any differences in experimental outcomes between male and female mice. To answer this question, we analyzed half-lives by sex. We did not note any effect of sex, at n=3 per sex per protein, no significant difference between male and female mouse terminal half-lives was detected. This analysis is now reported in the Results section, specifically in the “Pharmacokinetic analysis subsection”. 

5. The reviewer commented on the weight range of our mice, initially described as “between 25g and 35 g”. We have provided more detailed information, calculating a mean weight of 31 ± 4 g (mean ± SD, n=30) for all mice employed in this study, in the revised Methods section. We also stratified protein half-lives into two categories, heavier and lighter, and compared protein half-lives. At n=3 per weight class per protein, no significant difference between heavier and lighter mouse terminal half-lives was detected. This analysis is now reported in the Results section, specifically in the “Pharmacokinetic analysis subsection”, after the new statements concerning sex and half-life.

6. The reviewer asked if we used the same or different standards to quantify H6-MSA and H6-trC1INH(MGS). As stated in the Methods section, we used purified plasma-derived C1INH as a standard for the ELISA-based determination of H6-trC1INH(MGS) and H6-trC1INH(MGS)-MSA and we used purified recombinant H6-trC1INH(MGS)-MSA for the determination of H6-MSA. While it would arguably have been better to use the same standard for all 3 proteins, it should be noted that both ELISA protocols yielded identical sensitivities.

My co-authors and I are appreciative of the careful work of the editor and peer reviewers and have attempted to answer all queries and make all requested modifications to our work. We acknowledge that this revision has improved the scientific quality of our manuscript and hope that it is now acceptable for publication in PLOS ONE.

Sincerely,

William P. Sheffield, PhD

Professor, Pathology and Molecular Medicine, McMaster University

Hamilton, Ontario, Canada

---

## [Editor Report · Decision Letter 1]

5 Jun 2024

Prolonging the circulatory half-life of C1 esterase inhibitor via albumin fusion

PONE-D-24-10397R1

Dear Dr. Sheffield,

We’re pleased to inform you that your manuscript has been judged scientifically suitable for publication and will be formally accepted for publication once it meets all outstanding technical requirements.

Kind regards,

Yash Gupta, Ph.D.

Academic Editor

PLOS ONE
---

## [Editor Report · Acceptance letter]

13 Aug 2024

PONE-D-24-10397R1 

PLOS ONE

Dear Dr. Sheffield, 

I'm pleased to inform you that your manuscript has been deemed suitable for publication in PLOS ONE. Congratulations! Your manuscript is now being handed over to our production team.

Kind regards, 

on behalf of

Dr. Yash Gupta 

Academic Editor

PLOS ONE